# Carbothermic Reduction of Ilmenite Concentrate with Sodium Carbonate Additive to Produce Iron Granules and High Titania Containing Slag

**Zulfiadi Zulhan \*** , **Rifda Dinillah, Toto Yulianton, Imam Santoso and Taufiq Hidayat**

Metallurgical Engineering Department, Faculty of Mining and Petroleum Engineering,
Bandung Institute of Technology, Jl. Ganesa No. 10, Bandung 40132, Indonesia; rifdadinillah97@gmail.com (R.D.);
toto.yulianton@gmail.com (T.Y.); imam.santoso@metallurgy.itb.ac.id (I.S.); t.hidayat@itb.ac.id (T.H.)
**\*** Correspondence: zulfiadi.zulhan@itb.ac.id

**Abstract:** The influences of heating pattern and sodium carbonate addition on the carbothermic reduction of ilmenite concentrate have been experimentally studied. The experiments were carried out using isothermal–gradient temperature technique between 1000 °C and 1500 °C with different temperature profiles for a total reduction time between 110 and 160 min. The sodium carbonate was varied between 0 to 60 wt%. It was found that the temperature profile and sodium carbonate addition play an important role on the separation between metallic iron granule and titania rich slag. The optimum condition was achieved at initial and final reduction temperatures of 1300 °C and 1500 °C, respectively, with sodium carbonate addition of 30 wt%. At the optimum condition, the iron recovery was 97.1% and the solidified slag contained titanium pentoxide ($Ti_3O_5$), anatase ($TiO_2$), and sodium titanium dioxide.

**Keywords:** ilmenite; sodium carbonate; metal granules; titania rich slag

## 1. Introduction

Titania production steadily increased from 5.7 million tons in 2010 to 8.4 million tons in 2020 [1]. It was forecasted that titania production would continue to increase at a compound annual growth rate of 8.3% in 2021–2028 [2]. Titania has a wide range of applications in industries, such as for pigments, coatings, fillers in paper, rubber, plastics, textiles, inks, cosmetics, sunscreens, food additives, air purification, water purification, cancer treatment, biomedical, and photovoltaic [3–6]. Titania can be further processed to produce titanium sponges, titanium compounds, and titanium metal which are commonly used for applications in aircraft components, desalination plant, and electrical components. Approximately 95% of titanium is used in the form of oxide (titania) rather than in the form of metal or alloy.

Titania can be produced from ilmenite, rutile, anatase, brookite, titanite, and titano-magnetite. However, the main raw materials are ilmenite and rutile. Since the $TiO_2$ content of rutile is more than 95%, it can be fed directly to the chlorination process to produce titania or titanium chloride as a precursor for titanium metal production. Ilmenite can be treated either by leaching using the sulfate process to produce titania, or by combined carbothermic reduction using an electric furnace to produce pig iron and high titania slag ($TiO_2$ about 80%) which is further refined by the sulfate process to produce titania. The content of FeO in the slag is usually kept around 10–12% to maintain the fluidity of the slag thus ensuring good separation between metal and slag [7]. The high titania slag from the ilmenite smelting in an electric furnace is one of the main sources for producing titania, however its production requires high energy consumption since it takes place at a high temperature of about 1700 °C with a long retention time ranging from 8–10 h.

Therefore, an alternative method is needed to extract the titania from ilmenite at lower energy consumption, shorter treatment time, and more effective iron–slag separation.

The carbothermic reduction of ilmenite at 1000–1500 °C has been investigated by previous researchers. Guindy and Davenport [8] studied the reduction of synthetic ilmenite with graphite up to 1140 °C by thermogravimetric analysis and reported that the reduction was initiated at 860 °C at the graphite-ilmenite contact points. Gupta et al. [9] reduced synthetic ilmenite with graphite at 1000–1100 °C and observed that the reduction started at temperatures above 1000 °C. El-Tawil et al. [10] studied the kinetics of solid-state reduction of ilmenite ore by charcoal using self-reducing briquettes and found that a metallization degree of 97% was achieved at 1200 °C after a reduction time of 3 h. Coley et al. [11] reduced ilmenite using coal at 1314–1517 °C and produced carbon-saturated iron and titanium oxycarbide. Chen et al. [12] performed ball-milling of ilmenite and carbon mixture followed under vacuum at room temperature followed by reduction using thermogravimetric analysis apparatus. The ball milling treatment was found to enhance the subsequent reduction process and complete reduction of ilmenite to rutile and iron was achieved at 760 °C within 30 min. Welham and Williams [13] investigated the reduction of mechanically activated ilmenite with carbon and reported different titanium compounds were formed depending on the annealing temperature. Kucukkaragoz and Eric [14] investigated the solid-state reduction of natural ilmenite by graphite at 1250–1350 °C and reported that a reduction degree of over 80% was obtained at 1350 °C. Wang and Yuan [15] studied the reduction of ilmenite concentrate by graphite under argon atmosphere at 850–1400 °C using thermogravimetric analysis apparatus and found that the products of the ilmenite reduction were iron, $Ti_3O_5$, reduced rutiles, and pseudobrookite solid solution. Francis and El-Midany [16] carried out the reduction of ilmenite ore using carbon in a fixed-bed reactor at 1000–1200 °C under flowing nitrogen and reported that 99% metallization of iron was obtained from the reduction of ilmenite with a particle size of 200 mesh at 1200 °C within 6 h. Dewan et al. [17] performed reduction of an ilmenite concentrate using synthetic graphite at 1000–1500 °C under $H_2$, Ar, and He atmospheres. Under inert atmospheres of Ar and He, the reduction process occurred solely through the reaction between ilmenite and graphite. While under $H_2$ atmosphere, the reduction process was also influenced by the $H_2$ gas resulting in the ilmenite reduction which occurred at a lower temperature and at a faster rate. Pre-oxidation treatment of the ilmenite was suggested by several researchers to increase the reducibility of ilmenite [18–22].

The effect of additive on the carbothermic reduction of ilmenite at 1450 °C for 25–30 min was reported by Xu et al. [23]. Lithium carbonate ($Li_2CO_3$) of 2–3% and sodium sulfate ($Na_2SO_4$) of 7–8% were used as additives to produce slag with a $TiO_2$ content of 70–76% which can be separated from iron. Furthermore, Xu et al. [23] observed that the presence of sodium carbonate ($Na_2CO_3$) up to 7.5% together with sodium sulfate and lithium carbonate resulted in a more difficult iron and slag separation. Lv et al. [7,24,25] added 4% $Na_2SO_4$ to the ilmenite concentrate and found that an iron metallization degree of 94% and a slag containing 75% $TiO_2$ were obtained by a carbothermic reduction at 1450 °C for a reaction time of 30 min. Although the addition of $Na_2SO_4$ is found to be beneficial in the carbothermic reduction of ilmenite, the $Na_2SO_4$ can increase the sulphur content in the iron hence intensive desulphurization treatment would be required if the iron is to be used as feed in the steelmaking process. An alternative additive that can be considered is sodium carbonate which can eliminate sulphur pick-up in the iron product. El-Tawil et al. [26] investigated the addition of sodium carbonate to the carbothermic reduction of ilmenite at 1000–1200 °C. The iron metallization degree of 85% was achieved by the addition of 30 wt% sodium carbonate. The reduced iron was entrained in the slag and so crushing and grinding were required to liberate the metallic iron followed by magnetic separation.

In a previous report [27], the possibility of producing the metallic iron and $TiO_2$ rich slag by the carbothermic reduction of titanomagnetite-containing iron sand concentrate (59.18% Fe and 6.08% $TiO_2$) using an isothermal–gradient temperature technique was investigated. The characteristics of ilmenite are more or less similar to those of titanomagnetite.

In the present work, the combined effects of heating pattern and sodium carbonate additive on the separation of iron metal from the titania-containing slag in the carbothermic reduction of ilmenite have been experimentally investigated. The combined influences of these two important parameters on the ilmenite reduction have not been addressed systematically by the previous researchers.

## 2. Materials and Methods

The ilmenite concentrate was obtained from a tin concentrating plant in Bangka Island, Indonesia. The concentrate was ground using a ball mill for about 0.5 h to obtain a grain size of less than 100 mesh (−0.149 mm). The grain size distribution is shown in Figure 1. The concentrate was dried for 24 h at 130 °C. The chemical composition of the concentrate based on X-ray fluorescence (XRF) is provided in Table 1 which shows the presence of tin in the ilmenite concentrate. Coal was used as a reducing agent and was dried in an oven for 24 h at 130 °C. The coal was crushed and ground using ball mill to obtain a grain size of less than 65 mesh (−200 mm). The proximate, ultimate, and ash analyses of the coal are listed in Tables 2–4, respectively.

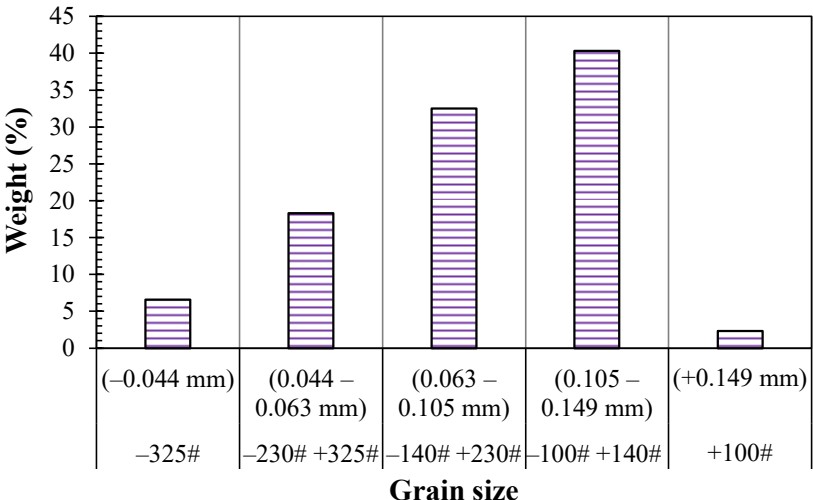

**Figure 1.** Grain size distribution of ilmenite concentrate.

**Table 1.** Chemical composition of ilmenite concentrate (in wt%).

| Ti | Fe | Mn | Sn | S | Cl | Si | Al | Nb | Y | Ta | W | Pb | O and Other |
|----|----|----|----|---|----|----|----|----|---|----|---|----|-------------|
| 33.23 | 21.21 | 2.03 | 1.88 | 0.98 | 0.67 | 0.65 | 0.64 | 0.62 | 0.32 | 0.29 | 0.23 | 0.17 | 37.08 |

**Table 2.** Coal proximate analysis (in wt%).

| Fixed Carbon | Ash | Volatile Matter | Inherent Moisture |
|--------------|-----|-----------------|-------------------|
| 44.63 | 12.20 | 40.00 | 3.17 |

**Table 3.** Coal ultimate analysis (in wt%).

| C | H | O | N | S |
|---|---|---|---|---|
| 65.77 | 5.34 | 14.21 | 1.19 | 1.29 |

**Table 4.** Coal ash analysis (in wt%).

| SiO$_2$ | Al$_2$O$_3$ | Fe$_2$O$_3$ | CaO | MgO | MnO | K$_2$O | Na$_2$O | SO$_3$ | Other |
|---------|-------------|-------------|-----|-----|-----|--------|---------|--------|-------|
| 47.03 | 24.88 | 14.29 | 3.76 | 1.15 | 0.22 | 0.52 | 0.25 | 3.81 | 2.95 |

The ilmenite concentrate was pre-oxidized in a muffle furnace at 1000 °C for 2 h. Three to four grams of pre-oxidized ilmenite concentrate were mixed with coal and sodium carbonate and then formed into a briquette with a diameter of 15 mm and a height in the range of 7.4 to 13.3 mm depending on the amount of additive. The weight of the pre-oxidized ilmenite concentrate was used as the basis for the addition of coal and sodium carbonate. The amount of coal in the briquettes was 10% and the addition of sodium carbonate was varied between 0 to 60%.

Each briquette was placed in a 20 mL porcelain crucible. A layer of coal was introduced between the briquette and the bottom of the crucible to prevent contact between the briquette and the crucible. Another layer of coal was added to cover the entire surface of the briquette to maintain a reducing environment. The total addition of coal for this purpose was 4 g. Finally, 1 g of alumina powder was added at the top of the coal layer to minimize the penetration of oxygen from the atmosphere during the reduction process. The experiments were conducted using the isothermal–temperature gradient profiles in a muffle furnace, and the temperature of the inside of the furnace was monitored with a type B thermocouple [27]. After the experiments, the crucibles were removed from the muffle furnace and the reduced briquettes were cooled naturally to room temperature in the crucibles. The reduced briquettes were weighed and documented. Three briquettes were prepared and reduced for each experimental condition. One of the reduced briquettes from each experimental condition was examined by Scanning Electron Microscope equipped with Energy-Dispersive X-ray Spectroscopy (SEM-EDS, JSM 6510 A, JEOL Ltd, Tokyo, Japan). Other briquettes were carefully crushed to enable the physical separation of metal granules from the slag. The metal granules were weighed, and their sizes were measured by a caliper. Finally, the solidified slags were analyzed by X-ray Diffraction (XRD, Smartlab, Rigaku, Tokyo, Japan).

## 3. Results and Discussion

### 3.1. Pre-Oxidation of Ilmenite Concentrate

Based on X-ray Diffraction (XRD) analysis in Figure 2, the minerals present in the as-received ilmenite concentrate were ilmenite ($FeTiO_3$), pseudorutile ($Fe_2Ti_3O_9$), rutile ($TiO_2$), and brookite ($TiO_2$). The chemical formula for ilmenite and pseudorutile can also be written as $FeO\ TiO_2$ and $Fe_2O_3\ 3TiO_2$, respectively. The pseudorutile appears to be an intermediate mineral between ilmenite and rutile which is formed due to the incorporation of $TiO_2$ and the oxidation of FeO in the ilmenite into $Fe_2O_3$.

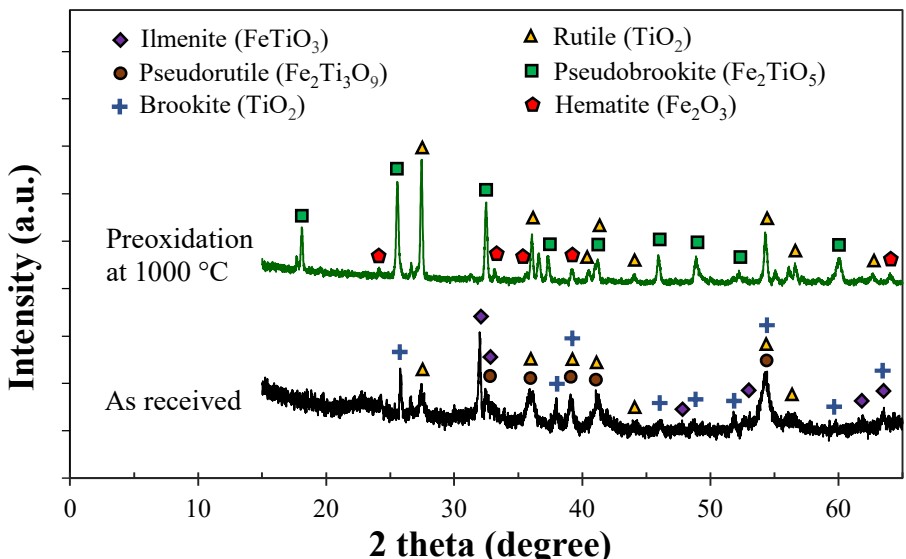

**Figure 2.** XRD spectra of ilmenite concentrate before and after pre-oxidation.

Previous researchers [18–22] suggested that pre-oxidation increase the reducibility of the ilmenite. The XRD analysis result of the pre-oxidized ilmenite concentrate at 1000 °C for 2 h in muffle furnace is shown in Figure 2. The following reactions may take place during the pre-oxidation:

$$2FeTiO_3 \text{ (ilmenite) (s)} + \frac{1}{2} O_2 \text{ (g)} = Fe_2O_3 \text{ (hematite) (s)} + 2TiO_2 \text{ (Rutile) (s)} \tag{1}$$

$$Fe_2Ti_3O_9 \text{ (pseudorutile) (s)} = Fe_2TiO_5 \text{ (pseudobrookite) (s)} + 2TiO_2 \text{ (Rutile) (s)} \tag{2}$$

$$Fe_2TiO_5 \text{ (pseudobrookite) (s)} = Fe_2O_3 \text{ (hematite) (s)} + TiO_2 \text{ (Rutile) (s)} \tag{3}$$

It can be observed in Figure 2, the ilmenite transformed into hematite and rutile based on the reaction in Equation (1). The pseudorutile in the concentrate transformed into pseudobrookite ($Fe_2TiO_5$) and rutile according to the reaction in Equation (2). The pseudobrookite was stable and did not dissociate into hematite and rutile as in reaction 3.

The experiment results in this research are in line with the results of previous researchers [18–20,22]. Merk and Pickles [18] performed the pre-oxidation on ilmenite in the air atmosphere at temperatures above 800 °C for at least 12 h and found that the ilmenite converted into pseudobrookite ($Fe_2TiO_5$). Similar results were reported by Gupta et al. [19] where the ilmenite was preheated under argon atmosphere at 1000 °C up to 2.5 h and transformed completely to pseudobrookite and rutile. Sun et al. [20] proposed a slightly different reaction scheme however with similar outcomes for the ilmenite oxidation in the temperature range from 800 to 1000 °C as follows:

$$4FeTiO_3 \text{ (s)} + O_2 \text{ (g)} = 4/3\, Fe_2Ti_3O_9 \text{ (s)} + {}^2/_3 Fe_2O_3 \text{ (s)} \tag{4}$$

Zhang et al. [22] investigated the pre-oxidation of ilmenite in the temperature range of 600 to 900 °C and reported that at temperatures less than 800 °C the ilmenite was oxidized to hematite and rutile (Equation (1)). At 900 °C, the reaction took place in two stages where in the first stage ilmenite was oxidized to hematite and rutile, and, in the following stage, rutile and hematite tended to combine into pseudobrookite, according to the reverse reaction of Equation (3).

## 3.2. Effect of Final Temperature and Sodium Carbonate Addition

The briquettes consisted of pre-oxidized ilmenite, 10% coal, and 20% $Na_2CO_3$ were reduced under the temperature profiles similar to that employed in the carbothermic reduction of titanomagnetite concentrate [27]. The temperature profiles are shown in Figure 3 where the initial temperature was 1000 °C and the final temperatures were 1300 °C for pattern A, 1400 °C for pattern B, and 1500 °C for pattern C. The holding time of the briquette at the initial temperature was 20 min. The heating rate was 10 °C/min from the initial to the final temperature at which the sample was kept for 30 min. The setting and measured temperatures according to the A, B, and C patterns referred to in the experiments were the inner temperatures of the furnace, not the temperatures in the crucible nor the surface of the briquettes.

No metal granule was observed in the reduced briquette cross section from the pattern A even with the addition of 20% $Na_2CO_3$ as shown in Figure 3a. Metallic iron may be formed, but the agglomeration of small metal particles into larger metal granules did not take place. Metal granules were clearly visible in Figure 3b,c from the reduced briquettes cross section from the pattern B and pattern C. The metal granules were formed inside the reduced briquettes which are different from the metal granules from the reduction of iron sand concentrate without additive [27] where granules of relatively small size were formed on the surface of the reduced briquettes. Higher final temperature promoted the agglomeration of small metallic iron particles into larger metal granules.

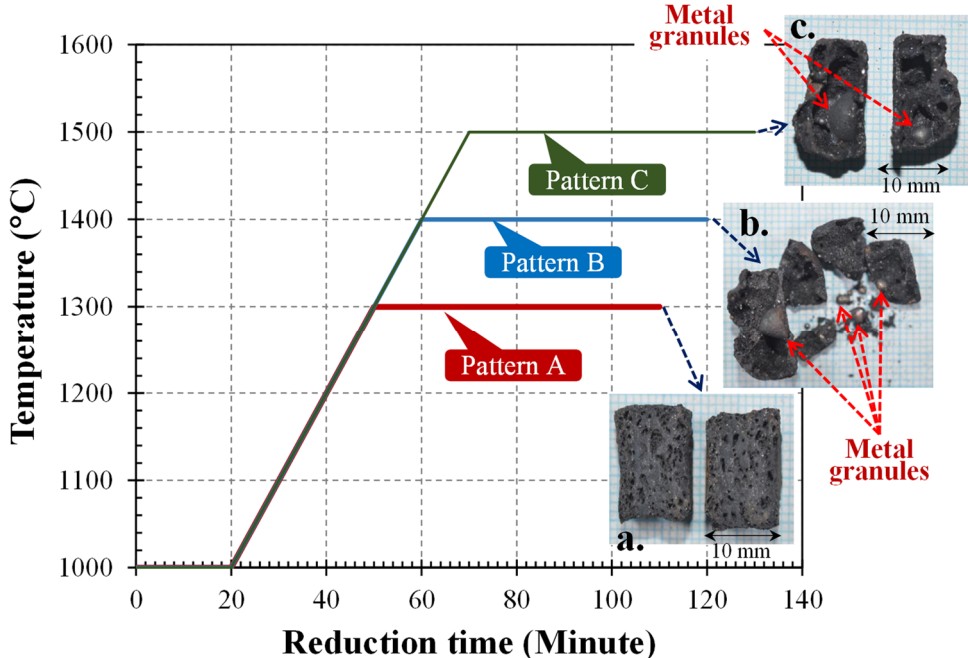

**Figure 3.** Temperature profiles and the physical appearances of reduced briquette cross section with the addition of 10% coal and 20% $Na_2CO_3$ ((**a–c**) are slag and metal composites).

The carbothermic reductions with 10% coal and variation of sodium carbonate between 0 to 30% have been performed and the results are summarized in Figure 4. Weight losses were observed in all samples which were combined outcomes of the removal of moisture and volatiles from coal, the dissociation of $Na_2CO_3$ into $Na_2O$ and $CO_2$, and the carbothermic reduction of oxides in the ilmenite concentrate. For the final temperature of 1300 °C (pattern A), although the decreases in weight were observed in all the reduced briquettes with 0 to 30 wt% $Na_2CO_3$ addition, the metal granules were only formed at 30% $Na_2CO_3$ addition. At the 30% $Na_2CO_3$ addition, the weight loss of briquette was 30.4%, the iron recovery in the granules was 52.3% (assuming 96% Fe content in the granules), the average granule size was 2.4 mm, and the number of granules was 4 pieces.

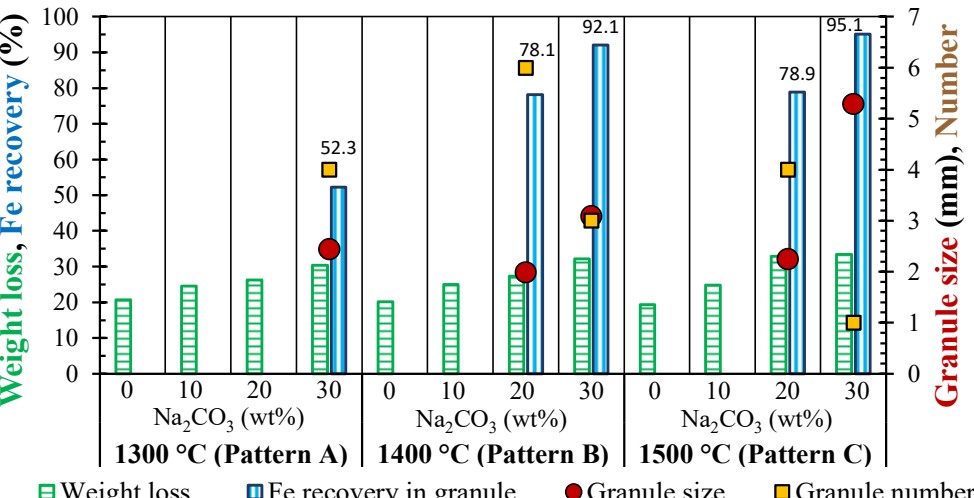

**Figure 4.** Effect of final temperature and addition of $Na_2CO_3$ on weight loss, iron recovery in granules, granule diameter, and number of granules from the carbothermic reduction of the ilmenite concentrate at different $Na_2CO_3$ addition, initial temperature of 1000 °C, different final temperatures, and heating rate of 10 °C/min.

As the final temperature increases to 1400 °C (pattern B in Figure 4), the metal granules already started to form in the briquette with the addition of 20 wt% $Na_2CO_3$. The increase of $Na_2CO_3$ addition from 20% to 30% increased weight loss from 27.3% to 32.2%, enhanced iron recovery in the granules from 78.1% to 92.1%, enlarged granule size from 2.0 to 3.1 mm, and reduced the granule number from 6 to 3 pieces. Similar to pattern B, pattern C indicated that metal granules had been formed with the addition of 20% $Na_2CO_3$. Increasing the addition of $Na_2CO_3$ from 20% to 30% in the briquette increased the weight loss from 32.9% to 33.4%, enhanced iron recovery in the granules from 78.9 to 95.1%, enlarged granule size from 2.2 to 5.3 mm, and reduced the number of granules from 4 to 1 piece. The experimental results indicated that the addition of $Na_2CO_3$ promoted the agglomeration of metals which resulted in the increasing size of metal granules and accordingly the decreasing number of metal granules.

The XRD spectra of the samples from the carbothermic reductions at a final temperature of 1400 °C (pattern B) with 10% coal and 0–30% $Na_2CO_3$ addition, and at a final temperature of 1500 °C (pattern C) with 10% coal and 20% $Na_2CO_3$ addition are shown in Figure 5. For a final temperature of 1400 °C (pattern B) without $Na_2CO_3$ and with 10% $Na_2CO_3$ additions, no metal granule was observed in the reduced briquettes, however, the XRD spectra of the metallic iron were detected in the samples as can be seen in Figure 5. The XRD spectra indicated that the reduction of the pre-oxidized ilmenite converted the pseudobrookite, rutile, and hematite in the samples into titanium pentoxide, anatase, and metallic iron. Crystalline sodium titanium oxide started to form in the reduced briquette with the addition of 10% $Na_2CO_3$. For reduction experiments at a final temperature of 1400 °C and 1500 °C (patterns B and C) with 10% coal and 20–30% $Na_2CO_3$ additions, the metal granules were formed as shown in Figure 4 and had been manually separated from the slags. It is worth to note that the XRD analysis only identified crystalline phases, i.e., titanium pentoxide, anatase, and sodium titanium oxide. The amorphous phase may also be present in the solidified slag and could not be detected by the XRD analysis.

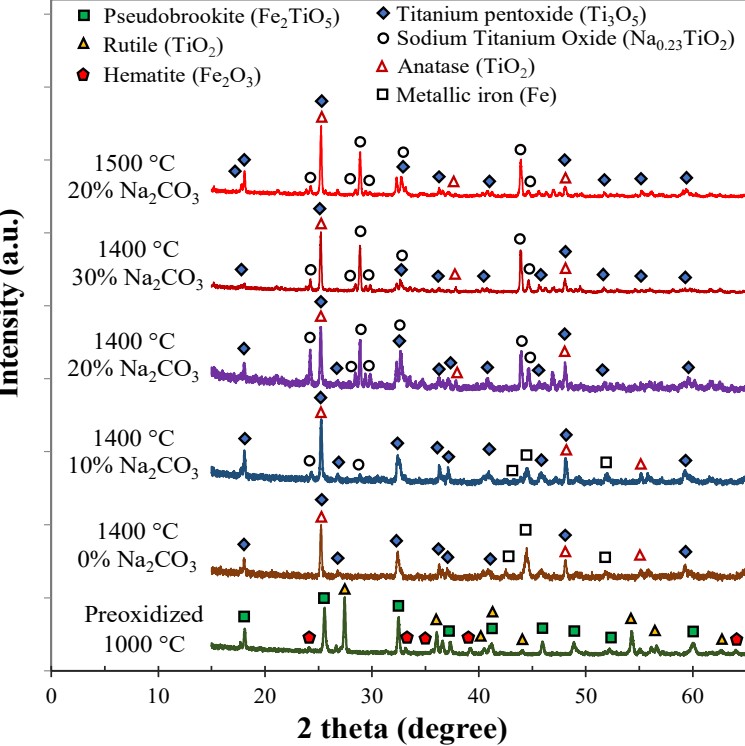

**Figure 5.** XRD spectra of the reduced briquettes for a final temperature of 1400 °C (pattern B) with 0% and 10% $Na_2CO_3$ additions, the slags after metal granules separation for a final temperature of 1400 °C (pattern B) with 20% and 30% $Na_2CO_3$ additions, and the slag after metal granules separation for a final temperature of 1500 °C (pattern C) with 20% $Na_2CO_3$ addition.

### 3.3. Effect of Initial Temperature

As shown in Figure 4, the highest final temperature of 1500 °C (the pattern C) provided good conditions to achieve high iron recovery in granules. Apart from the final temperature, the initial temperature may also affect the iron recovery in the metal granules. The carbothermic reductions of the ilmenite concentrates have been conducted using an isothermal–gradient temperature technique with varying initial temperatures as shown in Figure 6. The pattern D in Figure 6 is in principle similar to pattern C in Figure 3 which is the initial and final temperatures were 1000 °C and 1500 °C, respectively. The difference between the two is the heating rate where pattern C was 10 °C/min while pattern D was 6.6 °C/min.

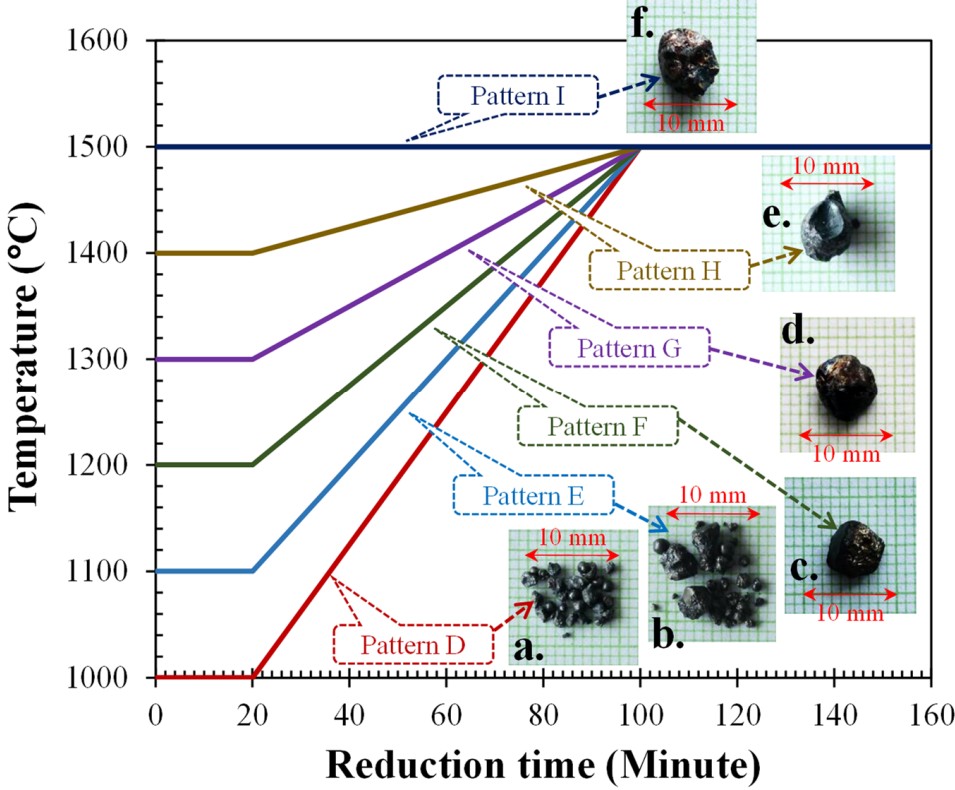

**Figure 6.** The temperature profile of ilmenite concentrate reduction by varying the initial temperature at 30% $Na_2CO_3$ addition ((**a**–**f**) are metal granules after separating from slag).

The result of initial temperature variation on weight loss, iron recovery in metal granules, granules size, and number of granules are summarized in Figure 7. Increasing the initial temperature from 1000 °C to 1300 °C (pattern D to pattern G) enhanced the iron recovery from 84.0% to 97.1%. Initial temperatures of 1400 °C (pattern H) and 1500 °C (pattern I) tended to lower iron recovery compared to the initial temperature of 1300 °C (pattern G).

The produced metal granules tended to agglomerate with increasing initial temperature from 1000 °C to 1200 °C as can be observed by the increasing granule size and the decreasing number of granules as shown in Figure 7. Only one granule was produced at an initial temperature between 1300–1500 °C with granule size around 5.0–5.6 mm. In summary, the experimental results in Figure 7 suggested that the pattern G (initial temperature of 1300 °C and final temperatures of 1500 °C) provided the optimum condition for increasing iron recovery in the metal granules.

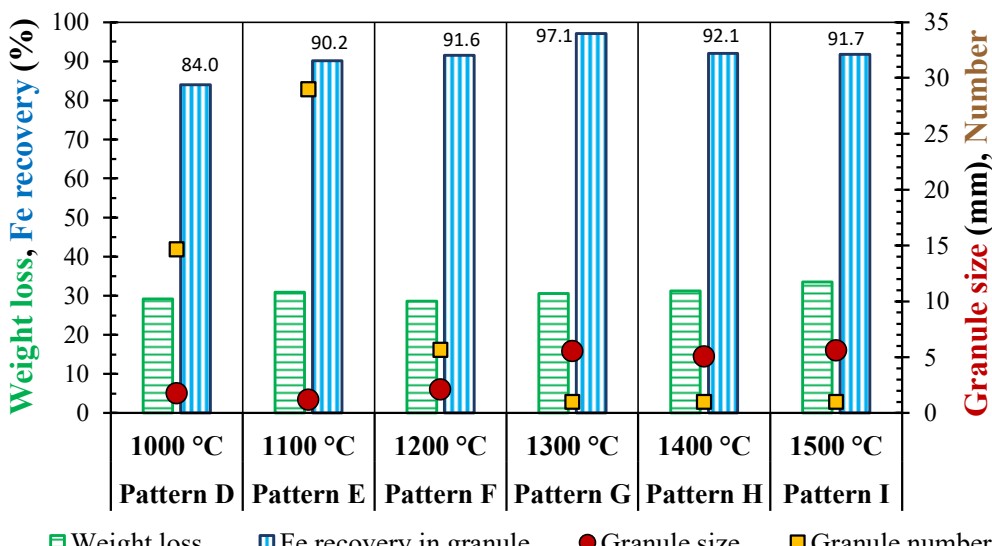

**Figure 7.** Effect of initial temperature on weight loss, iron recovery in granules, granule diameter, and number of granules from the carbothermic reduction of the ilmenite concentrate at 30% $Na_2CO_3$ addition, different initial temperatures, and final temperatures of 1500 °C.

### 3.4. Effect of Pre-Oxidation

As previously stated, the pre-oxidation facilitated the reduction of the ilmenite concentrate. The effect of pre-oxidation on the carbothermic reduction of the ilmenite concentrate with 30% $Na_2CO_3$ addition using pattern G (initial temperature of 1300 °C and final temperatures of 1500 °C) is shown in Figure 8. The experimental results verified that the pre-oxidation of the ilmenite ore increased iron recovery from 82.7% to 97.1%, enlarged the average granule size from 2.3 to 5.5 mm, and reduced granules number from 7 pieces to 1 piece.

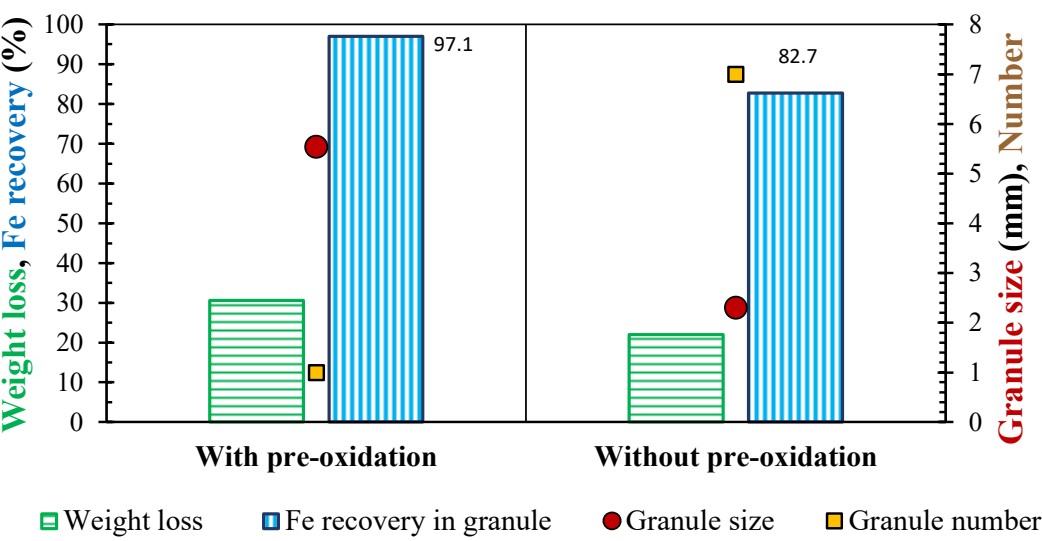

**Figure 8.** Comparison of briquettes with pre-oxidation and without pre-oxidation on weight loss, iron recovery in granules, granule diameter and number of granules from the carbothermic reduction of the ilmenite concentrate at 30% $Na_2CO_3$ addition, initial temperature of 1300 °C, final temperatures of 1500 °C, and heating rate of 6.6 °C/min (pattern G).

### 3.5. Effect of Sodium Carbonate Addition

The temperature profile of pattern G (initial temperature of 1300 °C and final temperatures of 1500 °C) was used to study the effect of sodium carbonate addition on the reduction of the ilmenite concentrate. The experimental results from the reduction experi-

ments using pattern G with 0–60% $Na_2CO_3$ addition are provided in Figure 9. Without the addition of sodium carbonate, iron granule was not formed. The absence of metal granule is similar to that observed in the experiments using the temperature profile of pattern C (initial temperature of 1000 °C and final temperatures of 1500 °C) with 0–10% $Na_2CO_3$ addition as indicated in Figure 4. Few metallic iron granules started to form at 15% $Na_2CO_3$ addition resulting in the iron recovery of only 3.2%. The average size of granules was 0.6 mm and the number of granules was 18 pieces. The experimental results suggested that the optimum $Na_2CO_3$ addition in the briquette was 30% since further addition did not give a significant increase in the iron recovery. Moreover, $Na_2CO_3$ addition above 30% in the briquette produced only one metallic iron granule with a relatively constant size at around 5.5 mm.

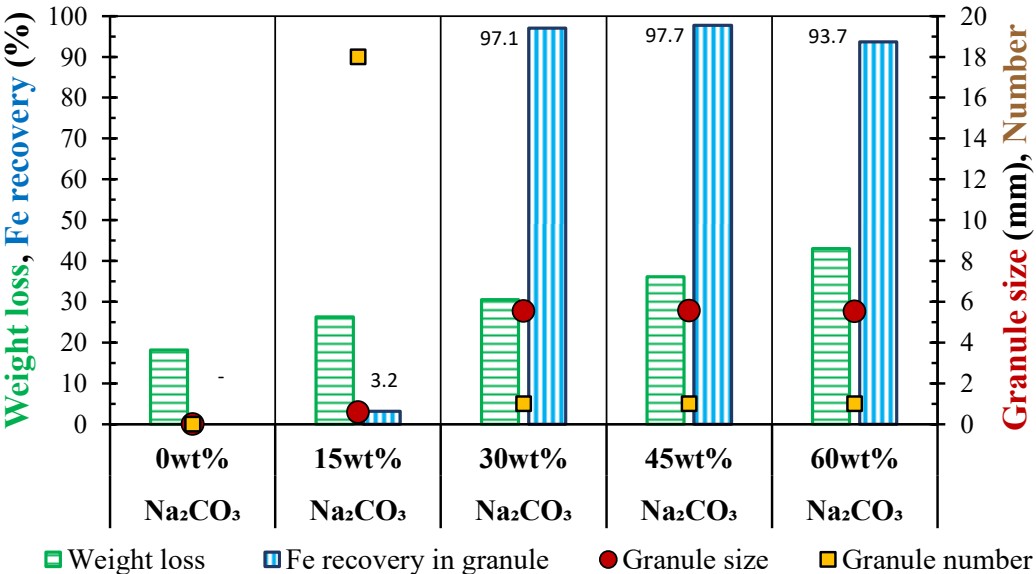

**Figure 9.** Effect of $Na_2CO_3$ addition on weight loss, iron recovery in granule, granule diameter, and number of granules from the carbothermic reduction of the ilmenite concentrate at different $Na_2CO_3$ additions, the initial temperature of 1300 °C, final temperatures of 1500 °C, and heating rate of 6.6 °C/min (pattern G).

Similar to the experimental result shown in Figure 4, the experimental results using the temperature profile of pattern G have also shown that the formation of metal granules was promoted by the $Na_2CO_3$ additive (Figure 9). During the carbothermic reduction of ilmenite, the $Na_2CO_3$ was first decomposed to $Na_2O$ and $CO_2$. The $Na_2O$ would then react with oxide components in the sample (such as $TiO_2$ and $SiO_2$) producing slag with relatively high liquid proportion as indicated in the ternary diagram of $Na_2O$-$TiO_2$-$SiO_2$ system [28]. The high liquid fraction in the slag facilitated the mobility of metals in the sample which led to the agglomeration of metals and the formation of metal granules. The formation of metal granules improved the separation of the metal product from the titania-rich slag and therefore increased the recovery of iron as shown in Figure 9.

The XRD spectra of the solidified slag after metal granules separation for 15%, 30%, and 60% $Na_2CO_3$ additions are shown in Figure 10. The crystalline phase detected in the slag was mainly titanium pentoxide ($Ti_3O_5$) or anatase ($TiO_2$). Another crystalline phase of sodium titanium oxide ($Na_{0.23}TiO_2$) was only detected in samples with 30% $Na_2CO_3$ addition.

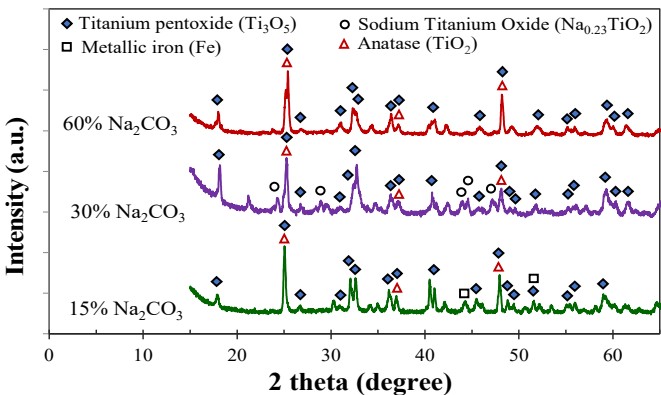

**Figure 10.** XRD spectra of slag after metal granules separation from the carbothermic reduction of the ilmenite concentrate at initial temperature of 1300 °C, final temperatures of 1500 °C, and heating rate of 6.6 °C/min (pattern G).

The elemental mapping using SEM-EDS was performed to obtain information on the distribution of Fe, Sn, Ti, Na, Si, Al, Mn, and S elements between phases in the samples and to acquire understanding on the separation of metallic iron and titania from other elements. The results of the elemental mapping of samples from the reduction experiments using heating pattern G with 15, 30 and 60% $Na_2CO_3$ addition are shown in Figure 11, Figure 12, and Figure 13, respectively. The iron oxides which were initially contained in ilmenite, pseudorutile, pseudobrookite, and hematite had been reduced to metallic iron. At 15% $Na_2CO_3$ addition, part of the metallic iron was in the form of fine prills which were entrained in the slag (Figure 11) and was detected by the XRD (Figure 10). The iron was separated from the titanium oxide as well as from other oxides which can be seen clearly in Figure 11b. Some of the tin oxide in the ilmenite concentrate was reduced to tin metal and dissolved into metallic iron solution (Figure 11c). Crystalline phase of titanium oxide/titania was formed and was clearly separated from sodium, silicon, and aluminum as depicted in Figure 11d–f. Most of the sodium combined with silicone, aluminum, and a small part of titanium to form an amorphous phase that was not detected by XRD spectra in Figure 10. Manganese tended to combine with sulphur around the metallic iron (Figure 11h,i).

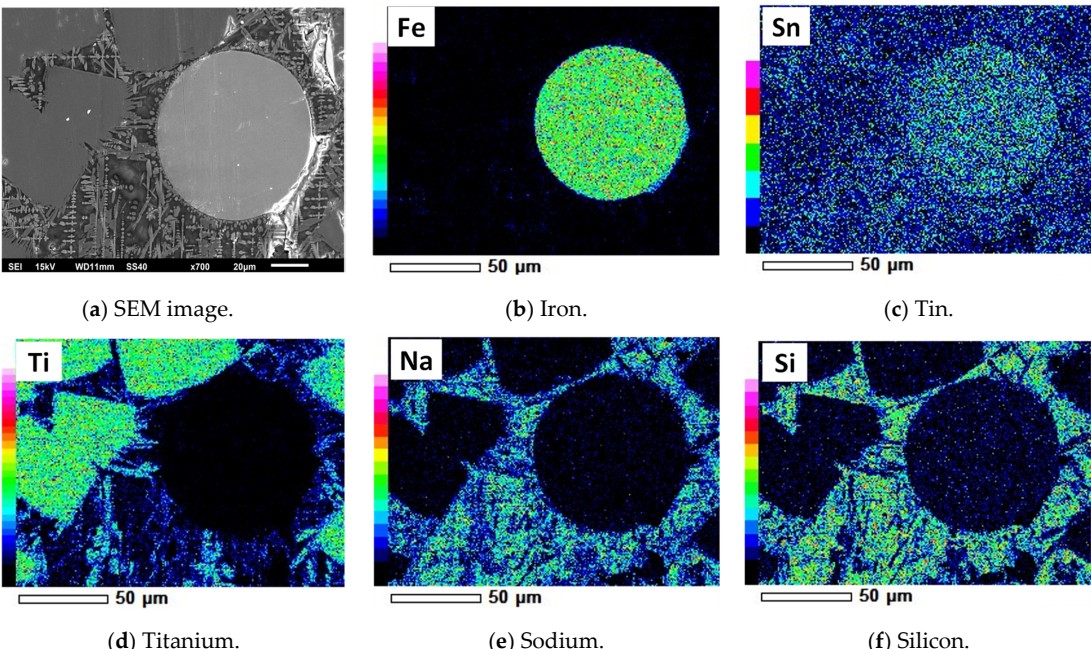

(**a**) SEM image.  (**b**) Iron.  (**c**) Tin.

(**d**) Titanium.  (**e**) Sodium.  (**f**) Silicon.

**Figure 11.** *Cont.*

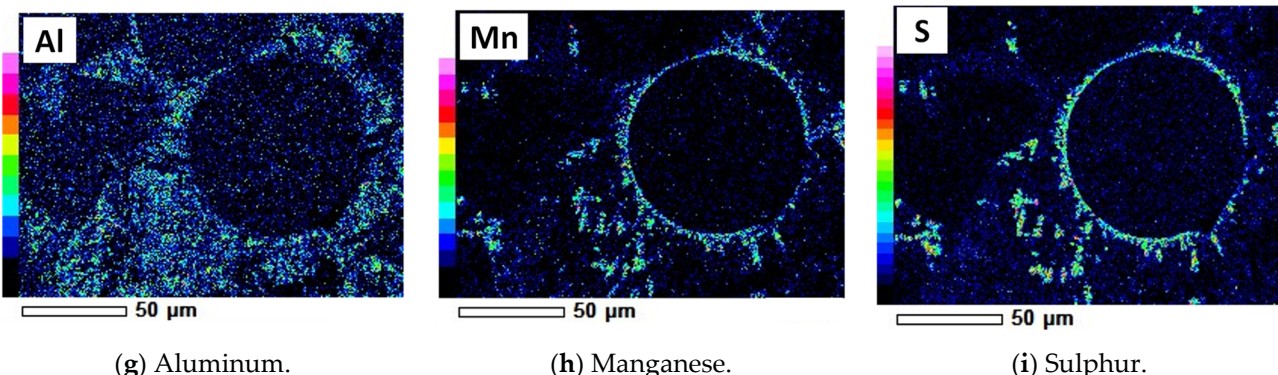

(**g**) Aluminum.　　　　　　(**h**) Manganese.　　　　　　(**i**) Sulphur.

**Figure 11.** SEM-EDS elemental mapping of slag from the carbothermic reduction of the ilmenite concentrate at 15% $Na_2CO_3$ addition, initial temperature of 1300 °C, final temperatures of 1500 °C, and heating rate of 6.6 °C/min (pattern G).

At 30% $Na_2CO_3$ addition, few metallic iron droplets with sizes less than 100 μm in the slag were physically entrained in the slag (Figure 12). The amount of the residual iron droplets in the slag was not significant as indicated by the high iron recovery in the form of granule (Figure 9) and the absence of metallic iron XRD spectra from the slag sample (Figure 10). Tin was distributed throughout the sample and some tin was concentrated in the iron droplets. Titania-rich crystals were also formed and were surrounded by the amorphous phase containing sodium, silicon, and aluminum with traces of manganese and sulphur. Association of sodium and titanium was observed in some area of the sample which indicates the presence of sodium titanate phase.

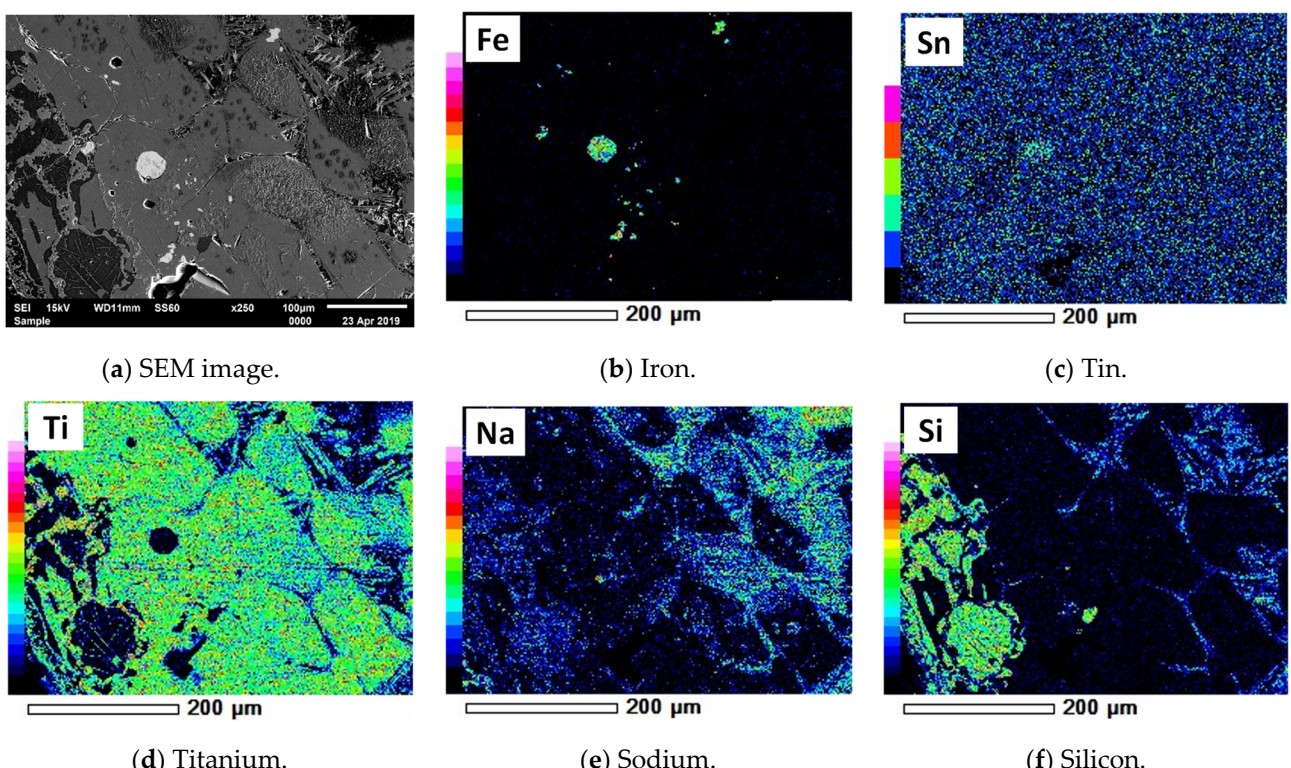

(**a**) SEM image.　　　　　　(**b**) Iron.　　　　　　(**c**) Tin.

(**d**) Titanium.　　　　　　(**e**) Sodium.　　　　　　(**f**) Silicon.

**Figure 12.** *Cont*.

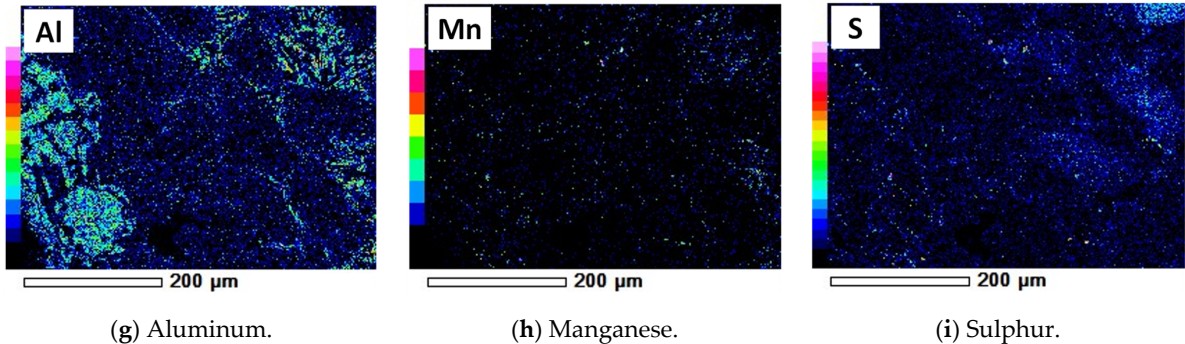

(**g**) Aluminum.                    (**h**) Manganese.                    (**i**) Sulphur.

**Figure 12.** SEM-EDS elemental mapping of slag from the carbothermic reduction of the ilmenite concentrate at 30% $Na_2CO_3$ addition, initial temperature of 1300 °C, final temperatures of 1500 °C, and heating rate of 6.6 °C/min (pattern G).

At 60% $Na_2CO_3$ addition, trace of residual iron droplets was also observed in the slag (Figure 13). Titania-rich crystal was observed in the forms of large crystal and several thin needle-like crystals. The amorphous phase contained dominantly sodium, silicon, and aluminum and had more homogeneous appearance.

(**a**) SEM image.                    (**b**) Iron.                    (**c**) Tin.

(**d**) Titanium.                    (**e**) Sodium.                    (**f**) Silicon.

(**g**) Aluminum.                    (**h**) Manganese.                    (**i**) Sulphur.

**Figure 13.** SEM-EDS elemental mapping of slag from the carbothermic reduction of the ilmenite concentrate at 60% $Na_2CO_3$ addition, initial temperature of 1300 °C, final temperatures of 1500 °C, and heating rate of 6.6 °C/min (pattern G).

It appears that the natural cooling of the slags obtained from the reduction experiments consistently resulted in the formation of high titania crystals as indicated by the XRD analysis and the elemental mapping using SEM-EDS. The formation of high titania crystals in the cooled slag is beneficial if physical separation of titania from other components is to be applied.

The overall slag composition at the optimum condition (heating pattern G, 30% $Na_2CO_3$ addition) was analysed by SEM-EDS area analysis. The analysis result showed that the slag contained 74% $TiO_2$ higher than that expected from the mass balance calculation (70% $TiO_2$) based on the amounts and compositions of the input materials. The difference between the measured and calculated $TiO_2$ content in the slag may be due to several factors such as, non-representative area selection for the SEM-EDS analysis, or volatilization of sodium at the reducing condition which resulted in the enrichment of $TiO_2$ in the slag. The titania-rich slag may be processed further to produce pure titania product, for example via the sulfate process.

The SEM microstructural and EDS compositional analyses of the iron granule from the reduction experiment using heating pattern G with 30% $Na_2CO_3$ addition are shown in Figure 14. The iron granule was consisted of iron phase and inclusions. The main iron phase contained around 95–97% Fe with around 2.10–2.61% Sn. There were two types of inclusions observed in the iron granule, i.e., iron–tin inclusion (point 2) and iron–manganese–sulfide inclusion (point 3). SEM-EDS analysis was performed on the overall iron granule area to measure its average composition. The iron granule had average composition of 91.4% Fe, 6.7% Sn, 1.04% Mn, and 0.78% S. Some extent of carbon dissolution in the iron granule is expected, however, the analysis of carbon content in the iron granule was not possible using the analytical tools available in the present study.

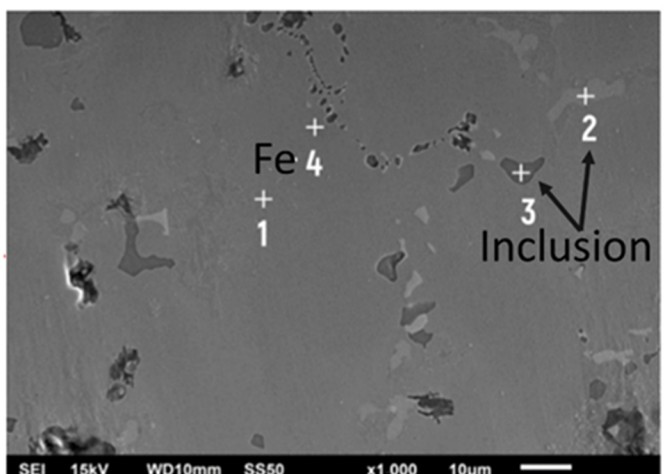

| Element | Position | | | |
|---|---|---|---|---|
| | 1 | 2 | 3 | 4 |
| Fe | 97.4 | 61.3 | 18.9 | 95.2 |
| Mn | 0 | 0 | 40.7 | 0.67 |
| S | 0 | 0 | 34.2 | 0 |
| Ti | 0 | 0 | 0 | 0 |
| Sn | 2.10 | 38.4 | 0.99 | 2.61 |

**Figure 14.** SEM EDS analysis secondary electron image and composition of iron granule from the carbothermic reduction of the ilmenite concentrate at 30% $Na_2CO_3$ addition, initial temperature of 1300 °C, final temperatures of 1500 °C, and heating rate of 6.6 °C/min (pattern G).

The ilmenite used in this study is a by-product of a tin concentration plant and hence tin contaminant in the ilmenite concentrate is expected. If the iron granule will be used as raw material for steel production, the sulphur in the iron–manganese–sulfide inclusion can be removed by fluxes addition during the steelmaking process. In the contrary, the tin in the iron granule will be difficult to remove during the steelmaking process. One of the common techniques that may be adopted to decrease the tin impurity content in the final steel product is by dilution using high-quality iron sources [29].

## 4. Conclusions

The experimental study on the carbothermic reduction of ilmenite concentrate has been performed. First series of experiments were conducted to investigate the effect of heating patterns on the ilmenite reduction. Experimental data from ilmenite reductions with different final holding temperatures and $Na_2CO_3$ additions at 10% coal addition, initial holding temperature of 1000 °C, and heating rate of 10 °C/min showed optimum condition at final holding temperature of 1500 °C and 30% $Na_2CO_3$ addition. Another experimental data from ilmenite reductions with different initial holding temperatures at 30% $Na_2CO_3$ addition, final holding temperature of 1500 °C, and total reduction time of 160 min showed optimum condition at initial holding temperature of 1300 °C. A second series of experiments was performed to investigate thoroughly the effects of pre-oxidation and sodium carbonate additive on the ilmenite reduction at the previously identified optimum heating pattern. It was found that pre-oxidation of ilmenite facilitated the reduction process and 30% $Na_2CO_3$ addition was sufficient to achieve optimum condition. At the optimum condition, the high iron recovery of approximately 97% was achieved due to the formation of iron granule with size of more than 5 mm which was favorable for its separation from the slag. It was also found that natural cooling of the slag led to the formation of titania-rich crystals in the form of titanium pentoxide ($Ti_3O_5$) or anatase ($TiO_2$) which may be further processed to produce high purity titania product. The present study demonstrated an alternative approach to the ilmenite reduction process with relatively low energy consumption, short processing time, and effective separation between iron metal and titania-rich slag.

**Author Contributions:** Conceptualization, T.Y., R.D. and Z.Z.; methodology, T.Y., R.D., Z.Z. and T.H.; software, Z.Z. and T.H.; validation, Z.Z. and T.H.; formal analysis, T.Y. and R.D.; investigation, T.Y. and R.D.; data curation, T.Y., R.D. and Z.Z.; writing—original draft preparation, T.Y., R.D. and Z.Z.; writing—review and editing, Z.Z., T.H. and I.S.; supervision, Z.Z. and T.H.; funding acquisition, Z.Z., T.H. and I.S. All authors have read and agreed to the published version of the manuscript.

**Funding:** This research was funded by P3MI–ITB 2020, Bandung Institute of Technology, Indonesia.

**Institutional Review Board Statement:** Not applicable.

**Informed Consent Statement:** Not applicable.

**Data Availability Statement:** Data is contained within the article.

**Acknowledgments:** The authors would like to thank PT. Timah Tbk, Indonesia, for providing the ilmenite concentrate.

**Conflicts of Interest:** The authors declare no conflict of interest.

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
