# Peer review of "Carbothermic Reduction of Ilmenite Concentrate with Sodium Carbonate Additive to Produce Iron Granules and High Titania Containing Slag"

_metals, doi:10.3390/met12060963_

Round 1
Reviewer 1 Report
The article is well written, with an adequate number of citations, although the literature review could be improved—for example, the last part of the first paragraph.
The procedure is straightforward, as are the results and discussions.
The explanation of SEM-EDS mapping can be improved (line 325). The authors include many pictures, but the analysis is weak.
The discussion about SEM-EDS microstructural and compositional analyses of the metal phase can be improved (line 342).
The article is excellent, it needs improvement in only a few aspects, but the conclusions are very poor. So, please, it is suggested to give more depth to the conclusions, highlighting the work that was done.
Author Response
Thank you very much for giving us the opportunity to revise our manuscript. We have carefully studied the reviewer's comments/suggestions. Comments/suggestions are constructive, and we have revised the manuscript accordingly. Please see our
point-by-point response to reviewers in the attached file below.

Reviewer 2 Report
The article presents the experimental results of a carbothermic reduction process of ilmenite concentrate with the addition of Na2CO3 at temperatures between 1000 °C and 1500 °C with different temperature profiles. The work was well designed and the result were properly presented. Minor revision is suggested. Some questions about this article are listed below.
- As mentioned in the paper, the process of carbothermic reduction of ilmenite with sodium carbonate additive has been reported previously, the novelty of this work should be more clearly addressed.
- What is the main role of sodium carbonate? A detailed explanation/discussion of the main role of sodium carbonate in section 3.5 is suggested.
- Fig. 9: Purity of Fe granules and TiO2 content of the slag should also be reported to give basic information based on which how the products can be further processed.
- Fig. 10: increasing Na2CO3 dosage from 15% to 30% led to the formation of Na0.23TiO2 phase and then it disappeared as Na2CO3 dosage increased from 30% to 60%, please explain.
- Fig.11-13: Na did not combine with Ti according to EDS. Please double check the formation of Na0.23TiO2 in the sample with 30% Na2CO3 dosage.
- The conclusions can be enhanced and improved by providing the implications of the results of the work.
Author Response

(The authors gave the same response as above.)
